# Extra Lateral Pin or Less Radiation? A Comparison of Two Different Pin Configurations in the Treatment of Supracondylar Humerus Fracture

**DOI:** 10.3390/children10030550

**Published:** 2023-03-14

**Authors:** Özgür Kaya, Batuhan Gencer, Ahmet Çulcu, Özgür Doğan

**Affiliations:** 1Department of Orthopedics and Traumatology, Faculty of Medicine, Lokman Hekim University, Ankara 06000, Turkey; 2Department of Orthopedics and Traumatology, Ankara City Hospital, Ankara 06000, Turkey; gencer.batuhan@gmail.com (B.G.); dr.ozgurdogan@gmail.com (Ö.D.); 3Department of Orthopedics and Traumatology, Ministry of Health Yüksekova State Hospital, Hakkari 30110, Turkey; dr.ahmetculcu@gmail.com

**Keywords:** supracondylar humeral fracture, pin configuration, radiation dose

## Abstract

Background: Closed reduction and percutaneous fixation are the most commonly used methods in the surgical treatment of supracondylar humerus fractures. The pin configuration changes stability and is still controversial. The aim of this study was to investigate the relationship between surgical duration and radiation dose/duration for different pinning fixations. Methods: A total of 48 patients with Gartland type 2, 3, and 4 supracondylar fractures of the humerus were randomized into two groups—2 lateral and 1 medial (2L1M) pin fixation (*n* = 26) and 1 lateral 1 medial (1L1M) pin fixation (*n* = 22). A primary assessment was performed regarding surgical duration, radiation duration, and radiation dose. A secondary assessment included clinical outcome, passive range of motion, radiographic measurements, Flynn’s criteria, and complications. Results: There were 26 patients in the first group (2L1M) and 22 patients in the second group (1L1M). There was no statistical difference between the groups regarding age, sex, type of fracture, or Flynn’s criteria. The overall mean surgical duration with 1L1M fixation (30.59 ± 8.72) was statistically lower (*p* = 0.001) when compared to the 2L1M Kirschner wire K-wire fixation (40.61 ± 8.25). The mean radiation duration was 0.76 ± 0.33 s in the 1L1M K-wire fixation and 1.68 ± 0.55 s in the 2L1M K-wire fixation. The mean radiation dose of the 2L1M K-wire fixation (2.45 ± 1.15 mGy) was higher than that of the 1L1M K-wire fixation (0.55 ± 0.43 mGy) (*p* = 0.000). Conclusions: The current study shows that although there is no difference between the clinical and radiological outcomes, radiation dose exposure is significantly lower for the 1L1M fixation method.

## 1. Introduction

Supracondylar humeral fractures are the most common elbow fractures in children, with incidences reaching a peak at about 5 to 8 years of age [1]. Generally, the fracture occurs due to a fall onto an outstretched hand, causing hyperextension of the elbow joint. The distal fragment is posteriorly displaced in more than 95% of fractures [2,3]. At the same time, these fractures have the potential to cause problems, especially due to elbow varus deformities, neurovascular injuries, stiffness, compartment syndrome, and malunion [4,5]. Therefore, the treatment of supracondylar humerus fractures is important in children because they affect function and elbow appearance.

According to Gartland’s criteria, these fractures are classified as nondisplaced fractures (type I), hinged fractures with the posterior cortex intact (type II), and completely displaced fractures (type III) [6], and in 2006, Leitch et al. [7] added type IV, which identifies fractures with multidirectional instability.

The most commonly used treatment for displaced supracondylar humeral fractures is percutaneous pinning after closed or open reduction. Due to the incision scar and long operation time, open surgery has more disadvantages than closed reduction and percutaneous pinning [8]. Although some studies found open reduction superior in terms of more satisfactory results, the general opinion is that supracondylar humerus fractures should be treated with closed reduction and percutaneous pinning [9,10]. The most common pinning configuration for supracondylar humeral fractures is cross-pinning or lateral pinning using two or three pins [4]. Although crossed medial–lateral pin fixation provides more stabilization than the lateral pin fixation method, it poses a greater risk of iatrogenic ulnar nerve damage [10,11,12,13]. In many biomechanical studies, the relationship between pin configuration and stability has been evaluated, and it has been shown that lateral pinning is more unstable against torsional forces than cross-pinning [14,15], and a third medial pin must be added whenever there is rotational instability [11,14,16,17].

In supracondylar humeral fracture surgery, the use of fluoroscopy is essential during the fixation phase. Intraoperative fluoroscopy is indispensable to orthopedic surgeons and boosts surgical accuracy. Therefore, both the surgeon and the patient are exposed to ionizing radiation during surgery. This ionizing radiation may increase the risk of developing cancer, which might be higher for pediatric patients [18,19]. Therefore, in supracondylar humeral fracture surgery, the methods that expose patients to less radiation should be chosen.

In our clinic, the surgical treatment of supracondylar humerus fractures generally requires the two lateral and one medial pin (2L1M) configuration, considering that this configuration is more stable, as has been proven in most biomechanical studies [16,17]. However, the prolongation of surgical time and the radiation dose received led us to wonder, “Is the second pin from the lateral side necessary?”. This led us to the conception of this study.

The aim of this study was to compare surgical duration, radiation duration, and the radiation dose regarding two different pin configurations (2L1M vs. 1L1M) for supracondylar humeral fracture.

## 2. Materials and Methods

The study design was a single-center, prospective, randomized clinical trial. Ethics approval was obtained from our institutional review board, and informed consent was provided by all of patients in the study. All study procedures were performed in accordance with the 1964 Declaration of Helsinki and all its subsequent amendments. Patients with Gartland type 2, 3, and 4 supracondylar humeral fractures who were treated (at our clinic) with closed reduction and percutaneous pinning were enrolled from April 2021 to December 2022.

All surgeries were performed by the same surgical team. The inclusion criteria were as follows: age: from 3 to 12 years old and the treatment of a displaced (type 2, 3, or 4) supracondylar fracture of the humerus. The exclusion criteria were as follows: an age of less than 3 years old or greater than 12 years old, an open fracture, a fracture requiring open reduction or neurovascular exploration, a floating elbow injury, and a bilateral upper extremity fracture. A total of 48 patients who met the criteria were included in the study (Figure 1). 

All patients underwent general anesthesia, closed reduction, and percutaneous pinning under the supine position [20]. The choice of pin configuration was determined according to the date of the day of surgery. If the date of the surgery fell on an odd day of that month (for example, the 1st, 3rd, or 5th day of the month), the first surgical method was chosen. If the operation date fell on an even day of that month (for example, the 2nd, 4th, or 6th day of the month), the second surgical method was chosen. With this method, patients were randomly differentiated. In the first method, following the closed reduction of the fracture, 2 lateral pins were placed under fluoroscopy in hyperflexion, and then 1 medial pin was placed by extending the elbow (2L1M group) (Figure 2). In the second method, following fracture reduction, 1 lateral pin was placed in hyperflexion under fluoroscopy, and 1 medial pin was placed by extending the elbow (1L1M group) (Figure 3). In both methods, the pins were bent outside the skin, and a bivalved, long-arm cast was applied with approximately 90° of elbow flexion and neutral forearm rotation. The duration of surgery, the duration of fluoroscopy used during surgery, and the amount of dose received were all recorded for both the reduction and fixation phases of the surgery. The radiation dose and duration were obtained through the recording system of the device used (Figure 4).

All of the patients were discharged after 1 or 2 days and were seen in the clinic 1 week after surgery. Radiographs were obtained in both anteroposterior and lateral planes at this follow-up. If these were acceptable, the child was seen again after 3 weeks, the cast was removed, and radiographs were obtained again. When acceptable healing was confirmed, the pins were removed in the clinic, and motion was encouraged. The mean immobilization time with pins in the present study was 5.2 ± 1.24 (4–7) weeks. Only those patients with elbow stiffness were routinely treated with physiotherapy.

Each patient was called for a control assessment every 3 months following pin removal. Clinical evaluation and radiographic evaluation were performed at each control visit. The clinical evaluation included the assessment of the carrying angle, the measurement of the passive range of elbow motion, a neurologic and vascular examination of the extremity, and the determination of any complications, such as superficial infection, deep infection, and the need for reoperation. The clinical results were graded according to the criteria of Flynn et al., which are based on the carrying angle and elbow motion [21]. The radiographic evaluation included an anteroposterior radiograph of the distal part of the humerus and a lateral radiograph of the elbow. A radiological assessment was made by comparing the Baumann angle, the humero-ulnar angle, and the humero-capitaller angle in the initial postoperative and final follow-up radiographs [4,22]. A change in Baumann’s angle of more than 12° was defined as a major loss of reduction, a change from 6° to 12° as mild displacement, and a change of less than 6° as no displacement.

The data obtained in the study were analyzed statistically using SPSS v. 22.0 software (IBM Corp., Armonk, NY, USA). Pearson’s chi-square and Fisher’s exact tests were used to compare the categorical data to independently assess the relationships among sex, age, Flynn’s criteria, and type of fracture. The Mann–Whitney U-test was used to compare the mean values between the groups.

## 3. Results

A total of 48 patients who met the inclusion criteria were operated on for displaced (type 2, 3, and 4) extension supracondylar humeral fracture. The mean age of the patients was 6.54 ± 2.02 (3–11) years. A total of 25 patients (52.1%) were female, with 12 patients (25%) having a Gartland type 2 supracondylar humeral fracture, 25 patients (52.1%) having a Gartland type 3 fracture, and 11 patients (22.9%) having a Gartland type 4 supracondylar humeral fracture.

A total of 26 patients were operated on using the first method (2L1M), and 22 patients were operated on using the second method (1L1M). There were 26 patients in the first group (2L1M), and the mean age of the patients was 6.57 ± 2.01 years. There were 11 (42%) females, and 6 patients (23%) had Gartland type 2 supracondylar humeral fractures, 11 patients (42%) had Gartland type 3 fractures, and 9 patients (35%) had Gartland type 4 supracondylar humeral fractures. According to Flynn’s criteria, 24 of the patients had excellent results, and 2 patients had good results.

There were 22 patients in the second group (1L1M), and the mean age of the patients was 6.51 ± 2.08 years. There were 14 (64%) females, and 6 patients (27%) had a Gartland type 2 supracondylar humeral fracture, 14 patients (63%) had a Gartland type 3 fracture, and 2 patients (10%) had a Gartland type 4 supracondylar humeral fracture. According to Flynn’s criteria, 17 patients had excellent results, and 7 patients had good results.

There was no statistically significant difference between the groups regarding age, sex, type of fracture, or Flynn’s criteria (Table 1).

Intraoperative radiographs showed that the fracture reductions were acceptable in all cases. A final clinical and radiological assessment confirmed the complete healing of the fractures in all children. There were no complications, such as iatrogenic ulnar nerve injury, a loss of reduction, pin tract infections, nonunions, etc., in either group.

The overall mean surgical duration for the 1L1M fixation (30.59 ± 8.72 min.) was statistically lower (*p* = 0.001) when compared to the 2L1M K-wire fixation (40.61 ± 8.25 min). The mean radiation duration was 0.76 ± 0.33 s. for the 1L1M K-wire fixation and 1.68 ± 0.55 s. for the 2L1M K-wire fixation. The mean radiation dose for the 2L1M K-wire fixation (2.45 ± 1.15 mGy) was higher than the mean radiation dose for the 1L1M K-wire fixation (0.55 ± 0.43 mGy) (*p* < 0.001). A comparison between the two types of fixations showed the mean surgical duration and radiation dose/duration were significantly lower for the 1L1M K-wire fixation compared to the 2L1M K-wire fixation (Table 2).

## 4. Discussion

Although closed reduction and percutaneous pinning are the primarily recommended treatment options in the literature for pediatric supracondylar humeral fractures, discussions continue regarding the superiority of different pin configurations (2L1M vs. 1L1M) to each other [4,9,10,11,12,13]. Although the main focus of these discussions is the absolute necessity of anatomical reduction and stabilization of the fracture, radiation exposure is an important criterion, especially considering that the patient population is in the pediatric age group [14,15,16,17,18,19]. The number of studies examining the superiority of different pin configurations to each other by including radiation exposure is quite limited in the literature, and this constitutes the main strength of our study. The most important finding of this study was that although there was no difference between the two different pin configuration methods in terms of clinical outcomes (*p* > 0.05 for each), there was a significant difference in terms of surgical duration, radiation dose, and radiation duration (*p* < 0.05 for each). Although the 1L1M pin configuration was clinically and radiologically similar, the radiation dose exposure it required was significantly lower (*p* < 0.001).

While closed reduction and percutaneous fixation are preferred in the surgical treatment of supracondylar humeral fractures in terms of low complication rates and wound healing, the pin configuration is still controversial [8,9,10,11,12,13,14]. Biomechanical studies show that cross-pinning is more effective than lateral pinning alone, especially to achieve greater rotational stability. In particular, it has been shown that stability is increased with the cross-pinning of two laterals and one medial [16]. When the literature was reviewed, the relationship between cross-pinning and lateral pinning in the percutaneous treatment of supracondylar humeral fractures was examined and evaluated in terms of stability and function. In particular, cross-pinning is considered to be more advantageous due to its contribution to torsional stability. The treatment of displaced supracondylar humeral fractures using only two lateral pins has been noted to be associated with a higher incidence of loss of reduction [23]. Although the medial pin improves torsional stability, it also introduces the risk of iatrogenic ulnar nerve injury. The reported incidences of postoperative ulnar nerve palsies range from 0% to 12% [24,25,26]. In a randomized clinical trial in 2007, Kocher et al. showed that both lateral entry pin fixation and medial and lateral entry pin fixation were effective in surgery for supracondylar humeral fractures [4]. They did not observe a functional difference in either group. They did not observe iatrogenic ulnar nerve injury in either group. In their surgical technique, firstly, the lateral pin was placed, the elbow was extended to a position of <90°, and a small incision was made over the medial epicondyle to protect the ulnar nerve. They believed that, with this method, the risk of ulnar nerve injury was reduced. In both groups, we applied medial pins to our patients, similar to the technique of Kocher et al. [4], following lateral pinning with an extension below 90 degrees, and we did not encounter any ulnar nerve injury either. Kocher et al. also suggested that a third pin may need to be administered to lateral entry patients, especially for those who are unstable. We believe that this recommendation is a method of increasing stability in accordance with the literature. In another prospective study conducted by Prashant et al. in 2016, medial–lateral cross-pinning was compared with lateral pinning, and they observed moderate reduction loss in two cases in the lateral pinning group [27]. While achieving similar results in terms of the functional and radiological aspects, they found iatrogenic ulnar nerve injury in two patients in the medial–lateral cross-pinning group. In the systematic review and meta-analysis study conducted by Dekker et al. in 2016 covering the years 1966–2015, it was observed that there was no significant difference between lateral pinning and medial–lateral cross-pinning [28]. Moderate reduction loss was seen in the lateral pinning group, and the medial–lateral cross-pinning group had a three-fold increase in iatrogenic ulnar nerve injury. In our study, we did not observe nerve injury or loss of reduction in any patients. In accordance with our study, it was observed that pin configuration did not have a significant effect on radiological or clinical outcomes (*p* < 0.05). At this point, it is obvious that there are many positive correlations, such as surgical experience and the effective use of fluoroscopy. We believe that having the same surgical team perform all the operations in our study facilitated optimal stability and explains the low complication rates. In addition, the fact that we did not include open fractures, bilateral injuries, or polytraumas may affect our complication rates. Finally, since the study mainly focused on perioperative variables, radiation exposure, and reduction quality, the fact that we did not evaluate our long-term follow-up results and changes in the patient’s joint range of motion and bearing angle may have also had an impact on our results. We can optimally compare the stability and functional and radiological results of the 1L1M and 2L1M configurations with further studies that also evaluate the mid-long-term follow-up results.

While the widespread use of radiography and even computed tomography in orthopedic pediatric injuries continues in diagnosis, treatment, and follow-up processes, the concept of limiting radiation exposure after traumatic injuries of pediatric patients has been a controversial issue in the literature for many years. Several studies have reported that with a correct and adequate physical examination of children with elbow injuries, the need for radiographic examination and radiation exposure could be significantly reduced [29,30,31]. Kraus and Dresing, in their 2023 study, investigated rational imaging in children and emphasized the necessity of protection from radiation [32]. Kocaoglu et al. investigated the necessity of fixation of both bones in children with distal forearm fractures of both bones, considering the surgical time and radiation exposure criteria, and reported that with the fixation of the distal radius fracture alone, optimal functional results could be achieved with less radiation exposure and shorter surgical time [33]. There are a limited number of studies in the literature comparing pin configurations and radiation doses in pediatric supracondylar humeral fractures. Patients with 18 supracondylar humeral fractures fixed by Martus et al. using the same method were shown to experience minimal exposure of radiosensitive organs to radiation doses [34]. Schmucker et al. determined the factors that influence radiation exposure during the fixation of supracondylar fractures [35]. No difference was found when they compared biplanar and uniplanar C-arm use. Both radiation exposure and duration increased with fracture displacement, and the number of pins increased. In the retrospective study conducted by Tzatzairis et al. in 2021, although there was no difference according to Flynn’s criteria, a statistically significant decrease in radiation dose was found in the cross-pinning group compared to the lateral pinning group [36]. They believed that the high radiation dose in the lateral pinning group was due to taking more images to assess stability. In our study, although there was no difference between the groups in terms of functional outcome, it was observed that the surgical time was longer and the radiation exposure was higher in the group with three pins (2L1M). We believe that after a safe level of stability of the cross-pin configuration has been achieved, the use of an extra pin may result in increased radiation exposure and prolonged surgical time.

Our study was limited in that only the radiation emitted through the C arm was evaluated, and this does not reflect direct radiation exposure. Although we did not study scatter radiation in our study, it is still an important factor to consider with regard to both the health of the patient and the surgical team. It would be appropriate to investigate this in future studies. Another important limitation was our relatively low number of patients. Finally, although we stated that the extra pin used in the 2L1M configuration did not contribute to stability while extending the surgical time and increasing the radiation exposure, as we mentioned before, our findings are supported by short-term follow-up results. The effect of the extra pin on stability and radiation exposure can be more clearly demonstrated by further studies with a larger patient cohort and including dosimetry analyses in which the mid-long-term follow-up results of the patients are examined.

## 5. Conclusions

In the surgical treatment of supracondylar humeral fractures, both 2L1M and 1L1M cross-pinning methods are effective. The 2L1M cross-pinning method prolongs the surgical time and causes greater radiation exposure. For this reason, we believe that when cross-pinning is applied in the treatment of supracondylar humeral fractures, the use of one lateral pin and one medial pin provides an effective result by minimizing radiation dose exposure. If persistent instability occurs after the placement of the crossed medial and lateral entry pins, the addition of a third pin is usually recommended in these patients to achieve fracture stability.

## Figures and Tables

**Figure 1 children-10-00550-f001:**
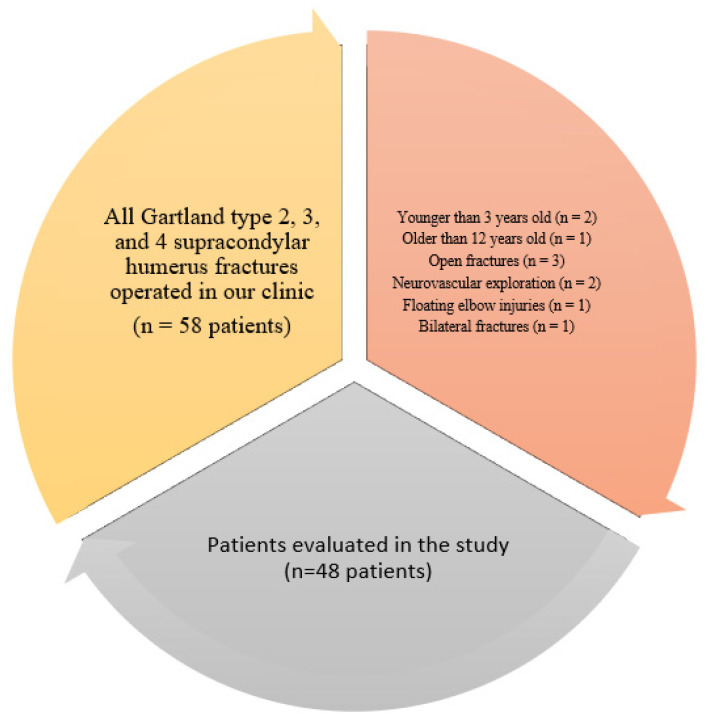
Detailed analysis chart of patients included and excluded from the study.

**Figure 2 children-10-00550-f002:**
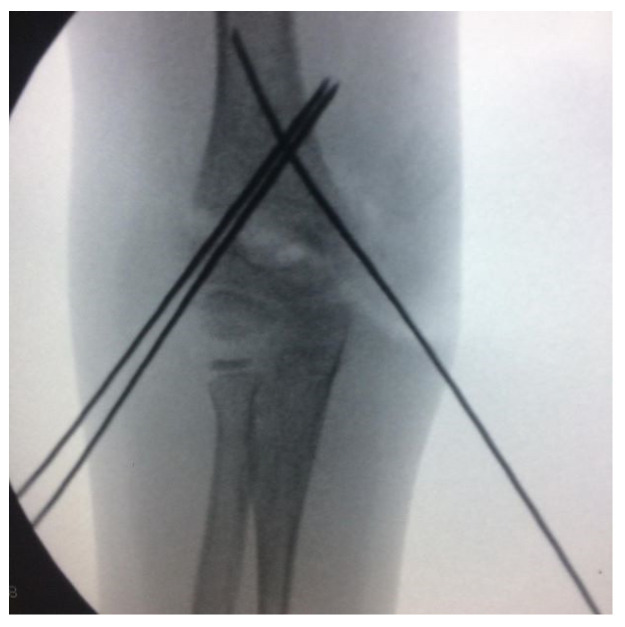
Two lateral and one medial pin configuration for supracondylar humeral fracture.

**Figure 3 children-10-00550-f003:**
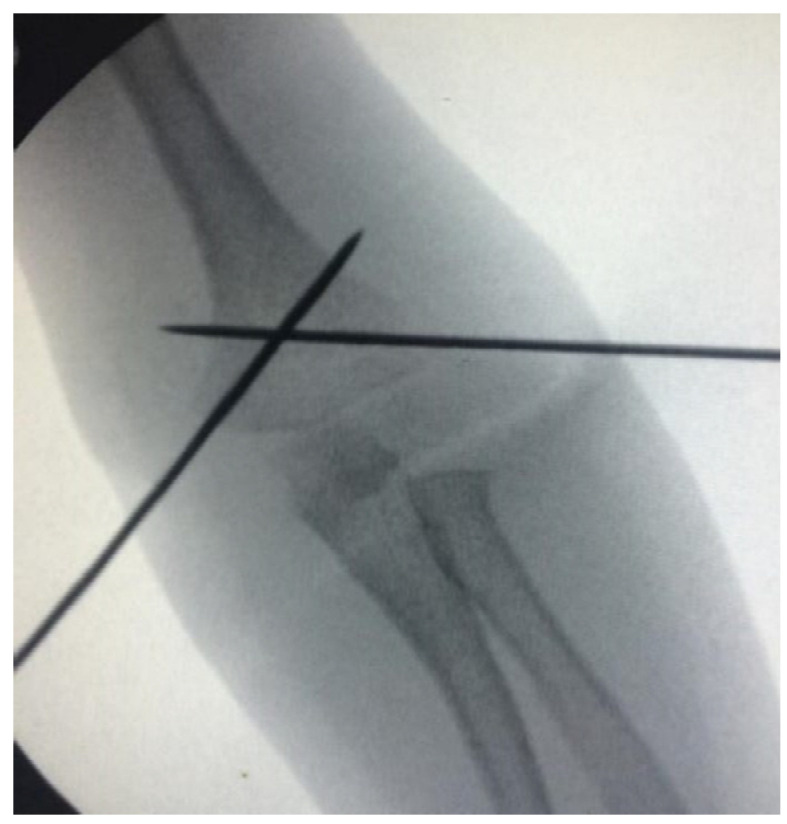
One lateral and one medial pin configuration for supracondylar humeral fracture.

**Figure 4 children-10-00550-f004:**
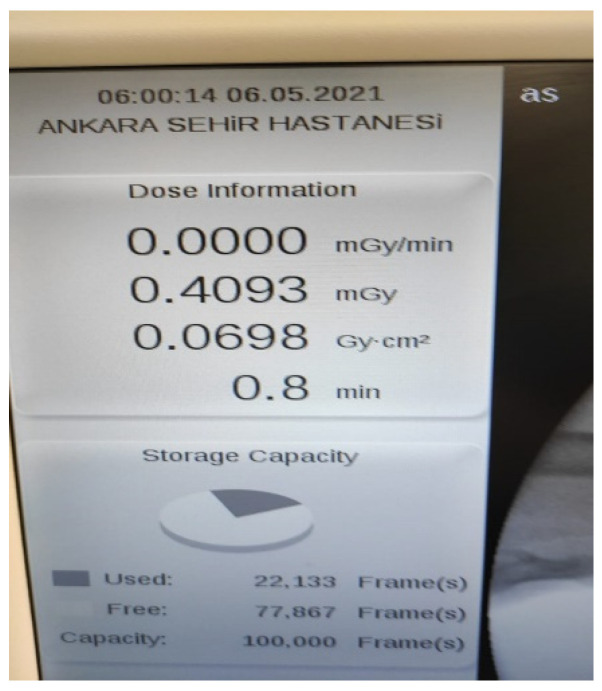
Radiation duration and dose recorded via fluoroscopy.

**Table 1 children-10-00550-t001:** Comparison of clinical results and descriptive features between two groups.

	2L1M	1L1M	*p* Value
No. of patients	26	22	
Age	6.57 ± 2.01	6.51 ± 2.08	0.925
Sex			0.141
male	15	8	
female	11	14	
Type of Fracture			0.105
Gartland 2	6	6	
Gartland 3	11	14	
Gartland 4	9	2	
Flynn Criteria			0.330
excellent	24	17	
good	2	7	
fair	0	0	
poor	0	0	

**Table 2 children-10-00550-t002:** Comparison of surgical time, radiation dose, and duration between pin configurations.

	Pin Configuration	N	Mean Rank	*p* Value
Surgical duration in min	2L1M	26	40.61 ± 8.25	
	1L1M	22	30.59 ± 8.72	
	Total	48		0.001
Radiation duration in min	2L1M	26	1.68 ± 0.55	
	1L1M	22	0.76 ± 0.33	
	Total	48		<0.001
Radiaton dose in mGy	2L1M	26	2.45 ± 1.15	
	1L1M	22	0.55 ± 0.43	
	Total	48		<0.001

## Data Availability

The data presented in this study are available on request from the corresponding author. The data are not publicly available due to privacy or ethical restrictions.

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
