# Peer review of "Extra Lateral Pin or Less Radiation? A Comparison of Two Different Pin Configurations in the Treatment of Supracondylar Humerus Fracture"

_children, 2023, doi:10.3390/children10030550_

Round 1
Reviewer 1 Report
It's interesting to know, how to reduce x-Ray dose
Author Response
Thank you for your evaluation. Professional language editing was made and certificate was added.

Reviewer 2 Report
line 159-160 - you mentioned that there was no statistical difference in flynn criteria between 1l1m and 2l1m groups, but in the table 1 p-value is 0,033 - less than 0,05 - how do you explain it?
Author Response
There was a typo in the table mentioned. Spelling error corrected. Sorry for the confusion.

Reviewer 3 Report
The Authors presented a very well written Paper about a very controversial matter in, probably, the commonest type of fracture in pediatric age. For that reason, I praise them for the significance of content (RCT trial) in particular to help surgeons on decision for the best kind of treatment.
I'd like to address just some minor revision:
-It's not clear if the duration of the surgery and the duration/amount of x-ray include the reduction phase or only the fixation phase. Please address it more clearly in the Paper.
-From line 178 you stated the surgical duration as 30.59 ± 8.72 sec in the first group and 40.61 ± 8.25 sec in the second (also reported in table 2). Just to be sure, is it really seconds and not minutes? If that so, do you consider the surgical duration only the fixing time? Please explain it and make it more clear in the Paper
-In the line 183 and in table 2, you reported a P-value of 0,000. I don't think is correct. Please explain it or correct it.
Author Response
- The Authors presented a very well written Paper about a very controversial matter in, probably, the commonest type of fracture in pediatric age. For that reason, I praise them for the significance of content (RCT trial) in particular to help surgeons on decision for the best kind of treatment.
- Thank you for your kind appreciation.
- It's not clear if the duration of the surgery and the duration/amount of x-ray include the reduction phase or only the fixation phase. Please address it more clearly in the Paper.
- We are sorry for the confusion. The time was recorded for both reduction and fixation phases. It was made clear in Materials and Methods Section.
- From line 178 you stated the surgical duration as 30.59 ± 8.72 sec in the first group and 40.61 ± 8.25 sec in the second (also reported in table 2). Just to be sure, is it really seconds and not minutes? If that so, do you consider the surgical duration only the fixing time? Please explain it and make it more clear in the Paper
- As mentioned in the above correction, the total time included both reduction and fixation phases and the time was measured as minutes not seconds. Sorry for the typo and confusion. Typo was corrected both on text and in table.
- In the line 183 and in table 2, you reported a P-value of 0,000. I don't think is correct. Please explain it or correct it.
- Sorry for the confusion. The P value was corrected as <0,001.
Reviewer 4 Report
The study was well conducted.
Linguistic editing should be done before publication.
In addition, for completeness in the discussion it would be appropriate to introduce the following bibliographical reference: (Wisdom M. et al (2023) "The role of the patient’s position in the surgical treatment of supracondylar fractures of the humerus: comparison of prone and supine position") be eligible for publication.
Author Response
- Moderate English changes required. Linguistic editing should be done before publication.
- Thank you for your evaluation. Professional language editing was made and certificate was added.
- The study was well conducted.
- Thank you for your kind appreciation.
- In addition, for completeness in the discussion it would be appropriate to introduce the following bibliographical reference: (Wisdom M. et al (2023) "The role of the patient’s position in the surgical treatment of supracondylar fractures of the humerus: comparison of prone and supine position") be eligible for publication.
- The mentioned article was cited in the Materials Methods section under citation number 20. Thank you for your contribution.
